# Prolonged secondary hyperparathyroidism in adenine-induced CKD leads to skeletal changes consistent with skeletal hyporesponsiveness to PTH

Corinne E. Metzger[1], Landon Y. Tak[1], Samantha Scholz[2], Matthew R. Allen[1,3,4]*

1 Department of Anatomy, Cell Biology and Physiology, Indiana University School of Medicine, Indianapolis, Indiana, United States of America, 2 Office for Research Compliance, Indiana University, Bloomington, Indiana, United States of America, 3 Department of Medicine, Division of Nephrology, Indiana University School of Medicine, Indianapolis, Indiana, United States of America, 4 Roudebush Veterans Administration Medical Center, Indianapolis, Indiana, United States of America

* matallen@iu.edu

**Data availability statement:** All relevant data are within the manuscript and its Supporting Information files.

**Funding:** This study was funded by the Department of Veterans Affairs via grants

## Abstract

High circulating parathyroid hormone (PTH) leading to secondary hyperparathyroidism is proposed to be a key driver of the skeletal phenotype of chronic kidney disease-mineral bone disorder (CKD-MBD) leading to high bone turnover and cortical bone deterioration. The association between high PTH and the skeletal phenotype is typically clearly demonstrated in preclinical models of CKD; however, clinical studies show the relationship between PTH and skeletal outcomes is not as clear. The clinical data have led to a proposed hyporesponsiveness to PTH in the CKD setting with unclear causes. In the current study, we assessed skeletally mature male C57BL/6J mice at 12-weeks and 21-weeks of adenine-induced CKD (Ad) with the second timepoint seven weeks longer than we have previously assessed. We found that serum BUN was high in Ad mice in both groups indicating the presence of kidney disease while PTH was higher in 21-wk Ad vs. 12-wk Ad. Despite the higher PTH, bone formation rate in 21-wk Ad mice was lower than 21-wk Ad mice. Additionally, immunohistochemical assessment of the PTH receptor, PTHR1, and RANKL, a key factor upregulated by PTH, showed a lower percentage of osteocytes positive for the proteins in 21-wk Ad vs. 12-wk Ad. Furthermore, regression analyses demonstrated a positive relationship between serum PTH and PTHR1 and RANKL at 12-weeks, but this relationship was lost by 21-weeks. Overall, these data indicate that prolonged exposure to continuously elevated PTH in adenine-induced CKD mice eventually led to an altered skeletal response indicating lower responsiveness of bone, particularly osteocytes, to the chronic PTH signal. This has implications for using PTH as a surrogate marker of bone outcomes in CKD as well as pointing to the need to better understand the time-based relationship between PTH and skeletal outcomes in CKD.

I01BX003025 and IK6-BX006479 awarded to MRA.

**Competing interests:** The authors have no competing interests to declare.

## Introduction

Chronic kidney disease (CKD) impacts approximately 10% of the world's population and is a leading cause of morbidity and mortality worldwide [1]. With declines in kidney function, complications impacting the skeleton occur, known collectively as CKD-mineral and bone disorder (CKD-MBD). Fracture rates are higher in patients with CKD compared to the general population at all ages [2] in both pre-dialysis [3–6] and dialysis patients [7–9]. Furthermore, post-fracture complications, including cardiovascular events, hospitalizations, and mortality, are high in CKD patients [9–11].

One component of CKD-MBD is high circulating parathyroid hormone (PTH) leading to secondary hyperparathyroidism which is postulated to cause high bone turnover with resorption exceeding formation favoring bone loss. For example, cortical bone deterioration over approximately ~1.5 years in CKD patients was related to circulating PTH levels [12]. However, clinical bone biopsy studies show a more complex picture with ~40–50% of CKD patients having low bone turnover, often termed adynamic bone disease in renal osteodystrophy [13]. One bone biopsy study found that the adynamic bone phenotype was associated with PTH levels above the upper limit of normal range [14], which contrasts with the idea that PTH is a potent stimulator of bone remodeling. Therefore, it appears the relationship between circulating levels of PTH and skeletal outcomes in clinical CKD is complex. For these reasons, skeletal hyporesponsiveness to PTH (also called skeletal resistance to PTH) has been proposed to occur in CKD [15], but what causes it and to what extent or in what conditions it occurs remains largely unknown.

In contrast to human clinical bone biopsy studies, pre-clinical animal models consistently demonstrate clear associations between high circulating PTH and cortical bone loss, particularly cortical porosity development [16–18], and high trabecular bone formation and bone resorption [18,19]. Animal models of CKD generally model a high PTH/high bone turnover phenotype including 5/6 nephrectomy [20], adenine-induced CKD [19], and genetic models like the Cy/+rat [21,22]. One challenge with translating animal models to clinical scenarios is that most animal experiments are relatively short duration. Therefore, the impact of prolonged (i.e., over 3 months) elevated PTH in an animal model of CKD is not well established. In this current project, we compared male C57BL/6J mice with adenine-induced CKD at two time points – 12-weeks and 21-weeks after initiating the adenine diet. We originally hypothesized that prolonged CKD with continuously high PTH levels at 21-weeks would lead to greater cortical porosity and higher bone turnover than 12-weeks. Our results contradict this hypothesis, suggesting this model, when allowed to continue development over time, may be suitable to study skeletal hyporesponsiveness to PTH.

## Methods

### Animals

Male C57BL/6J mice (n = 33) were purchased from Jackson Laboratories (JAX #000664) at 15 weeks of age and allowed to acclimate to the facility for one week. Mice were group housed 3–5 per cage in an institutionally approved animal facility

with 12-hr light/dark cycles and an average room temperature of 70° F. Treatments were based on cage due to the diet induction. Confounders due to cage location could not be fully controlled due to space within the facility. At 16 weeks of age, all control mice (CON; n = 15) were switched from the facility-provided grain-based diet to a purified casein-based diet with 0.9% phosphorous and 0.6% calcium (Teklad Diets [TD.150303]; Inotiv, Madison, WI, USA). All adenine-induced CKD mice (AD; n = 18) were switched to an identical purified casein-based diet with the inclusion of 0.2% adenine (Teklad Diets [TD.170948]; Inotiv). After a 6-week induction period on the diets, all AD mice were switched back to the control casein-based diet without adenine for the remainder of the study protocol as previously described [17,19]. The CON groups (n = 15) remained on the casein-based diet without adenine for the entire study protocol. After 12-weeks of adenine initiation, the 12-week groups (n = 9/group) were fully anesthetized via vaporized inhaled isoflurane and continuously monitored with the depth anesthesia confirmed via lack of toe pinch reflex. Animals were then humanely euthanized via exsanguination following by thoracotomy. The 21-week timepoint groups (CON = 6, AD = 9) were euthanized 21-weeks after the initiation of diets (Fig 1A). All mice were injected with fluorochrome calcein labels 7 and 2 days prior to euthanasia. All mice were checked daily for health, normal activity, and cage nest quality (described below) and body weight was monitored weekly. No animals showed signs of pain or distress or experienced rapid, unexpected weight loss throughout the protocol and no mice required supportive treatment of any kind. No analgesia was used or was necessary during the entire protocol. Previous work has established n = 6 as sufficient for detecting differences between control and adenine in serum PTH and cortical porosity. All animals made it to the planned study endpoint and no animals were excluded from analyses. All animal procedures were approved by the Indiana University School of Medicine Animal Use and Care Committee prior to the initiation of experimental protocols and methods were carried out in accordance with relevant guidelines and regulations.

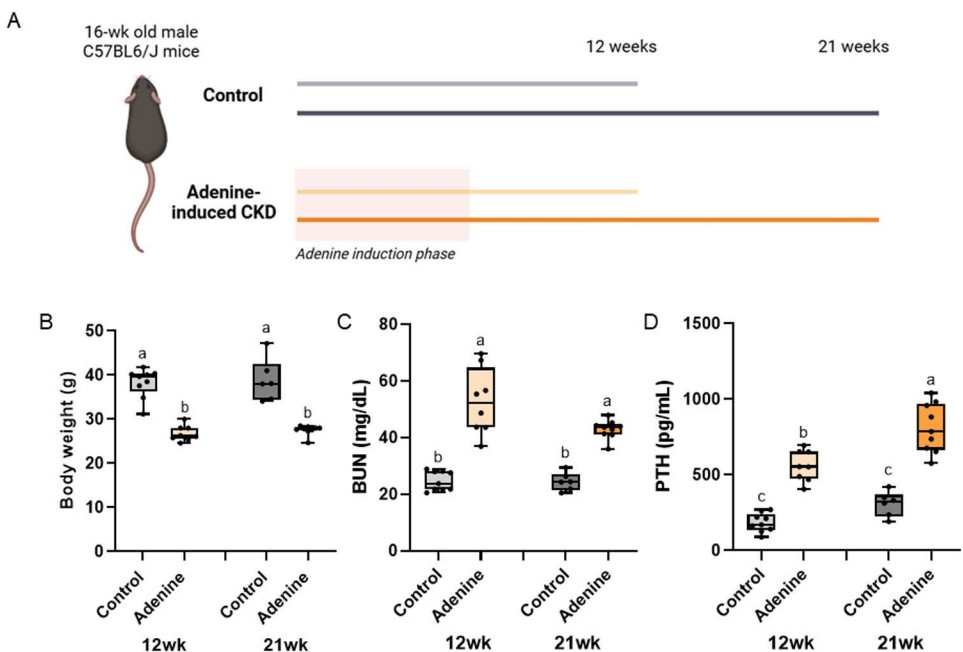

**Fig 1. Study schematic, body weight, and serum markers.** A) Schematic of study timeline. B) Body weight of adenine-CKD mice was lower than control regardless of time. C) Serum blood urea nitrogen was higher in adenine-CKD mice with no difference due to time. D) Serum PTH was highest in 21-week adenine mice followed by 12-week adenine mice with both control groups having the lowest PTH values. Bars not sharing the same letter are statistically different (p < 0.05).

## Cage enrichment and nesting scores

Due to weight loss while on the adenine diet, we provided mice with extra cage enrichment and examined nesting behavior within the cage to assess animal health and welfare as previously described [23]. Once a week, all cages received two Bed-r-Nest pucks (The Andersons, Inc., Maumee, OH, USA) and one Nestlet (Ancare, Bellmore, NY, USA) that were not separated/manipulated by caretakers. 24-hours later the cages were graded on the 3-dimensional nest (grades 2–5 with 2 a flat nest and 5 a full domed nest), intactness and incorporation of the Bed-r-Nest (scale of 1–5 with 1 untouched and 5 fully incorporated into nest), and intactness and incorporation of the Nestlet (scale of 1–5 with 1 untouched and 5 fully incorporated into nest). Individual cage scores were then combined to have a total cage score out of 15. Nesting scores were obtained from weeks 3–11 of the study protocol as this is the time that weight loss is most prominent in the model. The same nesting materials were provided to all cages for the entire duration of the study regardless of the scoring timeframe.

## Serum biochemistries

Cardiac blood collected at time of euthanasia was used to measure serum blood urea nitrogen (BUN) via colorimetric assay as an assessment of the presence of kidney disease (BioAssay Systems, Hayward, CA, USA). Serum 1–84 parathyroid hormone was measured via ELISA (Immnotopics Quidel, San Diego, CA, USA) with an intra-assay coefficient of variation of 2.4–5.6% and an inter-assay coefficient of variation of 5.5%. All serum assays were run in duplicate with serum samples that had not been through any freeze/thaw cycles and followed all the procedures of the manufacturer.

## Micro-computed tomography

The right distal 1/3 femora were scanned, 3 at a time in a multi-bone holder, as previously described [24] on a SkyScan 1272 system (Bruker, Billerica, MA, USA) with a 0.5 aluminum filter and an 8 μm voxel size. Scans were reconstructed and rotated to a standard orientation utilizing SkyScan software (NRecon, Dataviewer). Trabecular bone was analyzed in a 1 mm region starting proximal to the growth plate in the distal femur utilizing CtAn at a binary threshold of 110–255. Cortical bone properties were analyzed in a 1 mm region starting ~2.5 mm proximal to growth plate in the distal femur. Cortical porosity was assessed from hand drawn regions of interest (tracing the periosteal and endosteal surfaces) and measured with an inverse binary threshold of 110–0. In addition, % cortical porosity, average pore size, and number were assessed. Cortical bone area was measured from the same region of interest, but analyzed with a binary threshold of 110–255. Finally, cortical thickness was measured with a flooded threshold (0–255) to obtain an average width from the endocortical to periosteal edges without accounting for porosity within the region.

## Histomorphometry

Following micro-CT scanning, fixed undemineralized right distal femurs were serially dehydrated and embedded in methyl methacrylate (Sigma Aldrich, St. Louis, MO). Serial frontal sections were cut 4 μm-thick and left unstained for analysis of fluorochrome calcein labels. For trabecular measures, a standard region of interest of trabecular bone excluding primary spongiosa and endocortical surfaces was utilized. Total bone surface (BS), single-labeled surface (sLS), double-labeled surface (dLS), and interlabel distances were measured at 20x magnification. Mineralized surface to bone surface (MS/BS; [dLS+(sLS/2)]/BS*100), mineral apposition rate (MAR; average interlabel distance/5 days), and bone formation rate (BFR/BS; [MS/BS*MAR]*3.65) were calculated. Intracortical analyses were completed on both adenine-induced CKD groups (control animals without cortical pores cannot be analyzed) within standardized regions that utilized both cortices (approximately 600 mm² of tissue analyzed). The number of labeled pores, total labeled surface within pores, and interlabel distance were obtained. Calculations included pore number/bone area, mineral apposition rate (interlabel distance/5 days) and bone formation rate [100*MAR*(label length/2)/bone area].

A second 4 μm-thick frontal section of the distal femur was stained with tartrate resistant alkaline phosphatase (TRAP) for assessment of osteoclasts. For trabecular bone, sections were analyzed for osteoclast-covered trabecular surfaces normalized to total trabecular bone surface (Oc.S/BS, %) within the same trabecular region of interest as described above. For intracortical analyses within the two adenine groups, the osteoclast surface per bone surface within cortical pores (Oc.S/Pore Surface, %) was obtained.

A final 4 μm-thick section was stained with Von Kossa/McNeal for assessment of trabecular osteoid surfaces (OS/BS, %) and osteoid thickness (O.Th). These measurements were obtained in the same trabecular region of interest as described above. Histomorphometric analyses were performed using BIOQUANT (BIOQUANT Image Analysis, Nashville, TN) and each analysis was performed by a single individual to eliminate inter-individual variability. All nomenclature for histomorphometry follows standard usage [25].

### Immunohistochemistry

The right proximal half of the femur was decalcified in 14% EDTA for ~2 weeks. Samples were subsequently embedded in paraffin and longitudinally sectioned to 5 μm thickness. Sections were stained utilizing a standard avidin-biotin method. Samples were stained for receptor activator of nuclear facto κB ligand (RANKL; Abcam, Cambridge, MA, USA), para-thyroid hormone receptor-1 (PTHR1; Abcam), and annexin V, an early cellular marker of apoptosis (Abcam). Peroxidase development was performed with an enzyme substrate kit (DAB, Vector Laboratories, Burlingame, CA, USA) and nuclear stain counterstaining with methyl green (Vector Laboratories). Negative controls for all antibodies were completed by omitting the primary antibody. Analyses were completed in both the trabecular bone for the proximal femur and in the cortical bone closest to midshaft. Results are reported as the percentage of osteocytes stained positively for the protein (DAB-positive) relative to all osteocytes (DAB-positive and methyl green-positive) in the regions of interest. All analyses were completed by the same individual.

### Statistical analyses

All data were assessed for normality using Shapiro-Wilk. Nesting scores and body weight across the adenine induction and early recovery phase were measured by a mixed model ANOVA with time as a repeated factor. If data were normally distributed, data were analyzed with a 2x2 Factorial ANOVA (disease-by-time) with all main and interaction effects recorded. When the model 2x2 ANOVA p-value was statistically significant, a Tukey HSD post hoc test was completed to assess individual group differences. When data did not meet normality assumptions (with a Shapiro-Wilk test of $p < 0.05$), data were run with a Kruskal-Wallis test. When the Kruskal-Wallis test was statistically different, a Steel-Dwass post hoc test was completed to assess individual group differences while controlling for the overall error rate. For pore network from micro-CT and pore histology assessments, which were only done in CKD groups due to the presence of porosity, normally distributed data were analyzed with a Student's t-test. Data that did not meet normality assumptions were analyzed with a Wilcoxon test. Data that did not meet normality standards are called out with the nonparametric test used in the results text (Kruskal-Wallis or Wilcoxon). Regression analyses were completed between serum PTH (independent factor) and cortical osteocyte PTHR1 and RANKL (separately as dependent factors). Adjusted $R^2$ values and p-values are reported for regression analyses. All data are presented as mean ± standard deviation with all individual data points shown in graphs. All statistical analyses were completed on JMP Pro 17 (SAS, Cary, NC, USA).

## Results

### Final body weight was lower in adenine groups compared to control groups regardless of time point but nesting scores did not differ

During and following the adenine induction period, adenine-induced CKD mice had lower body weight than controls at all time points ($p < 0.0001$). Nesting scores, however, were not different across groups, except at week 7, when adenine

cages had a higher nesting score than control cages (p = 0.0489; S1 Fig). Body weight was statistically different at the study end points (Kruskal Wallis p < 0.0001). Adenine mice at both 12 and 21 weeks had lower bodyweight than both control groups with no differences due time (Fig 1B).

### All adenine groups had high serum BUN while 21-week adenine mice had the highest serum PTH

Serum BUN showed a statistical difference (Kruskal-Wallis p < 0.0001) with both adenine groups having levels higher than both controls groups regardless of time (Fig 1C). Serum PTH also showed a statistical difference (p < 0.0001) with a main effect of both disease (p < 0.0001) and time (p < 0.0001), but no interaction effect (p = 0.1215). 21-week adenine had the highest PTH followed by 12-week adenine. Both control groups were lower than both adenine groups (Fig 1D).

### Trabecular bone volume was lower in adenine groups regardless of time

The factorial ANOVA for trabecular bone volume showed an effect of disease (p < 0.0001), but no effect of time (p = 0.4365) and no interaction effect (p = 0.8571). Both adenine groups had lower trabecular bone volume than both control groups (Fig 2A). Similarly, trabecular thickness showed an effect of disease (p = 0.0024), but no time effect (p = 0.3700) and no interaction effect (p = 0.8084). Trabecular thickness was statistically lower in 12-week adenine vs. 21-week control (Fig 2B). Trabecular separation showed statistical difference (p < 0.0001) with 21-week adenine having the highest value followed by 12-week adenine and then 12-week control (Fig 2C). Trabecular number also showed statistical difference (p = 0.0009) with both adenine groups lower than the time-matched control group (Fig 2D).

### Bone formation rate was higher in 12-week adenine compared to all other groups

Bone formation rate failed normality tests and was run with a Kruskal-Wallis (p = 0.0005). 12-week adenine had higher bone formation rate compared to all other groups while 21-week adenine was not different from control groups (Fig 3A). Mineralized surface (MS/BS, Fig 3B) showed only an effect of time (p = 0.0037) with no effect of disease (p = 0.1395) and no interaction effect (p = 0.3601). Mineral apposition rate showed an effect of disease (p < 0.0001) and time (p = 0.0158), but no disease-by-time interaction (p = 0.0758). MAR was higher in both adenine groups compared to control groups with 12-week adenine higher than 21-week adenine (Fig 3C). Osteoid trabecular-covered surfaces showed statistical differences (p = 0.0035) with both adenine groups higher than both control groups (Fig 3D).

### Osteoclasts were higher in both adenine groups compared to both control groups

Osteoclast-covered trabecular surfaces showed statistical difference (Kruskal-Wallis p < 0.0001) with both adenine groups higher than both control groups (Fig 3E). Osteoclast number showed a main effect of disease (p < 0.0001), but no effect of time (p = 0.2972) and no disease-by-time interaction (p = 0.9933). Both adenine groups were higher than both control groups (Fig 3F).

### Cortical bone area was lower and cortical porosity was higher in adenine groups compared to control groups

Cortical bone area showed a main effect of disease (p < 0.0001), but no effect of time (p = 0.2979) and no interaction effect (p = 0.2486). Both adenine groups had lower cortical bone area than both control groups regardless of time (Fig 4A). There were no statistical differences in total bone area (p = 0.1433), marrow area (p = 0.9740), or cortical thickness (Kruskal Wallis p = 0.0840; Fig 4B). Cortical porosity showed statistical difference (Kruskal Wallis p < 0.0001) with both adenine groups higher than both control groups (Fig 4C). When comparing volumetric pore analyses between both adenine groups, overall cortical porosity was higher in 21-week adenine vs. 12-week adenine (p = 0.0052; Fig 5A). Additionally, 21-week adenine had greater average length of pores (p = 0.0293; Fig 5B) and higher pore number (p = 0.0170; Fig 5D) than 12-week adenine. There were no differences in average length between pores (Wilcoxon p = 0.8946; Fig 5C). Between 12-week

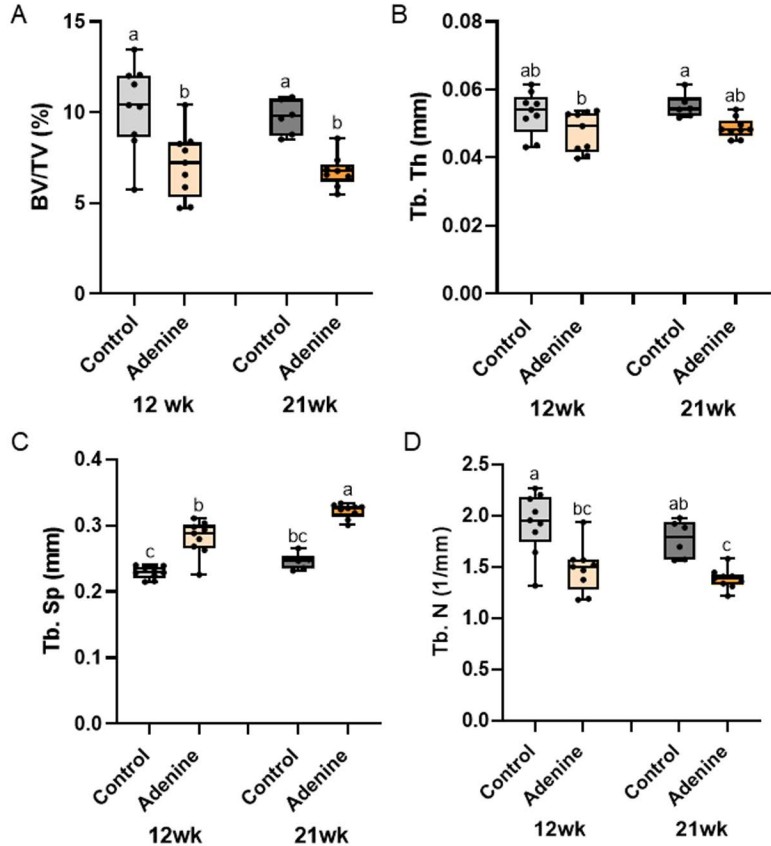

**Fig 2. Trabecular microarchitecture of the distal femur.** A) Trabecular bone volume (BV/TV) was lower in both adenine groups compared to both control groups. B) Trabecular thickness was statistically lower in 12-week adenine vs. 21-week control with the other groups not statistically different from any group. C) Trabecular separation was higher in both adenine groups compared to their age-matched control group. D) Trabecular number was lower in both adenine groups compared to their age-matched control group. Bars not sharing the same letter are statistically different (p<0.05).

and 21-week adenine, there were no differences in labeled pores (p=0.3048; Fig 6A), mineral apposition rate (p=0.2479; Fig 6B), bone formation rate (p=0.9345; Fig 6C), or osteoclast-covered pore surfaces (p=0.4348; Fig 6D).

**PTHR1- and RANKL-positively stained osteocytes were both higher in 12-week adenine mice compared to 21-week adenine mice**

There were significant differences in PTHR1-positive trabecular osteocytes (Kruskal-Wallis p<0.0001). 12-week adenine had the highest group average followed by 21-week adenine; both of which were higher than controls. Within cortical bone, 12-week adenine levels were higher than all other groups (Kruskal-Wallis p<0.0001) with no difference between adenine and control at the 21-week time point (Fig 7A).

Trabecular bone RANKL-positive osteocytes had a main effect of disease (p<0.0001) and a disease-by-time interaction effect (p=0.0002), but not a statistically significant main effect of time (p=0.0655). 12-week adenine had the highest group average followed by 21-week adenine and then both control groups. Within cortical bone, 12-week adenine levels were higher than all other groups (Kruskal-Wallis p=0.0001) with no difference between adenine and control at the 21-week time point (Fig 7B).

For annexin V-positive osteocytes, there was a main effect of disease (p<0.0001), but no effect of time (p=0.0864) nor an interaction effect (p=0.1929) in trabecular bone. At both time points, adenine groups were higher than both control

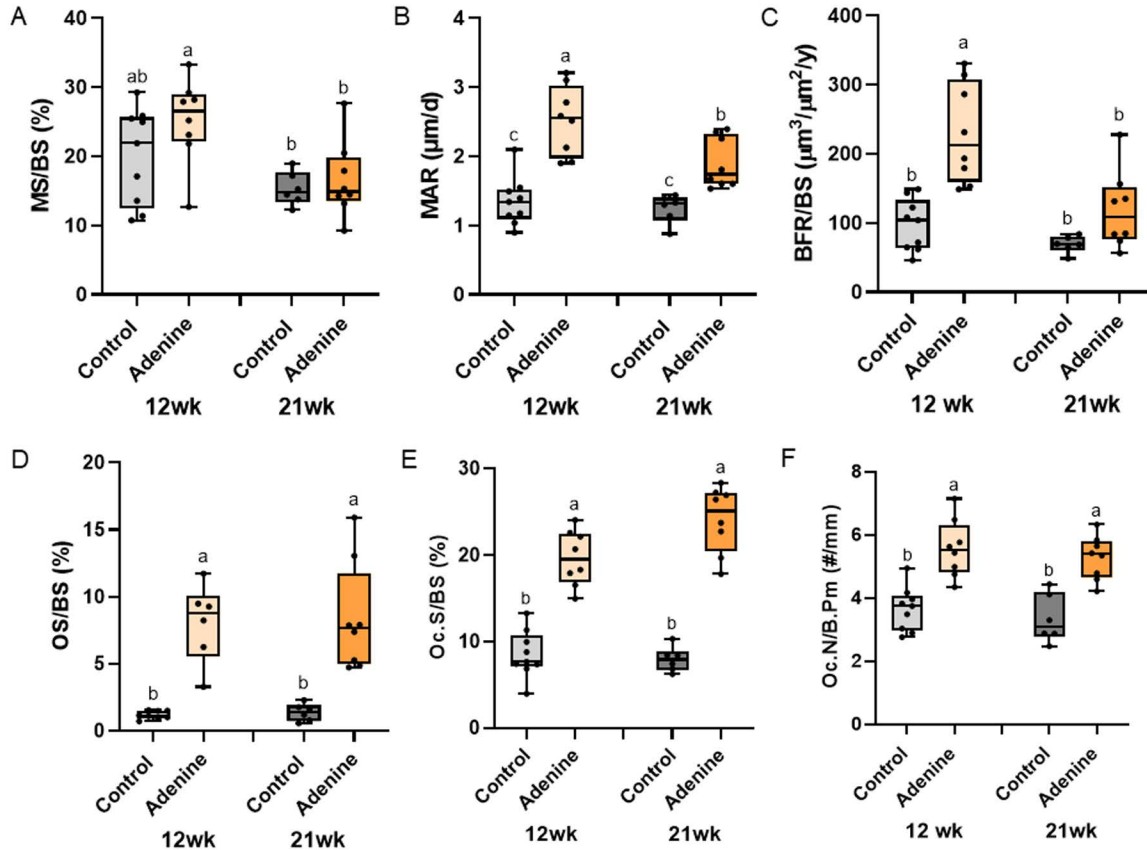

**Fig 3. Histomorphometry of the distal femur trabecular bone.** A) Mineralized surfaces (MS/BS) were lower in both 21-week groups compared to 12-week adenine. B) Mineral apposition rate was higher in both adenine groups compared to both control groups with 12-week adenine higher than 21-week adenine. C) 12-week adenine had the highest bone formation rate (BFR/BS) of all groups. D) Osteoid-covered trabecular surfaces were higher in both adenine groups compared to both control groups. E) Osteoclast-covered trabecular surfaces were higher in both adenine groups vs. both control groups regardless of time. F) Osteoclast numbers were higher in both adenine groups with no difference due to time. Bars not sharing the same letter are statistically different (p < 0.05).

groups. Within cortical bone, there was a main effect of disease (p < 0.0001), but no effect of time (p = 0.5051). There was a disease-by-time interaction (p = 0.0290) with 12-wk adenine higher than 12-wk control, but 21-wk adenine not statistically different than 21-wk control (Fig 7C).

**Serum PTH statistically predicts ~60–70% of the variability in osteocyte PTHR1 and RANKL in the 12-week group, but not the 21-week group**

Regression analysis between serum PTH and cortical osteocyte PTHR1 across both time points showed no statistical significance ($R^2$ = 0.0561, p = 0.1995; Fig 8A). Within the 12-week time point, circulating PTH did statistically predict ~71% of the variability in cortical osteocyte PTHR1 ($R^2$ = 0.7165, p < 0.0001; Fig 8B). At the 21-week time point, there was no statistical significance ($R^2$ = 0.1919, p = 0.1025; Fig 8C).

Regression analysis between serum PTH and cortical osteocyte RANKL at both 12- and 21-weeks showed no statistical relationship ($R^2$ = 0.0002, p = 0.9315; Fig 8A). Within only the 12-week time point, serum PTH statistically predicted ~60% of the variability in cortical osteocyte RANKL ($R^2$ = 0.6111, p = 0.0003; Fig 8B). Assessing only the 21-week time-point, there was a mild relationship between serum PTH and cortical osteocyte RANKL ($R^2$ = 0.2617, p = 0.0485; Fig 8C),

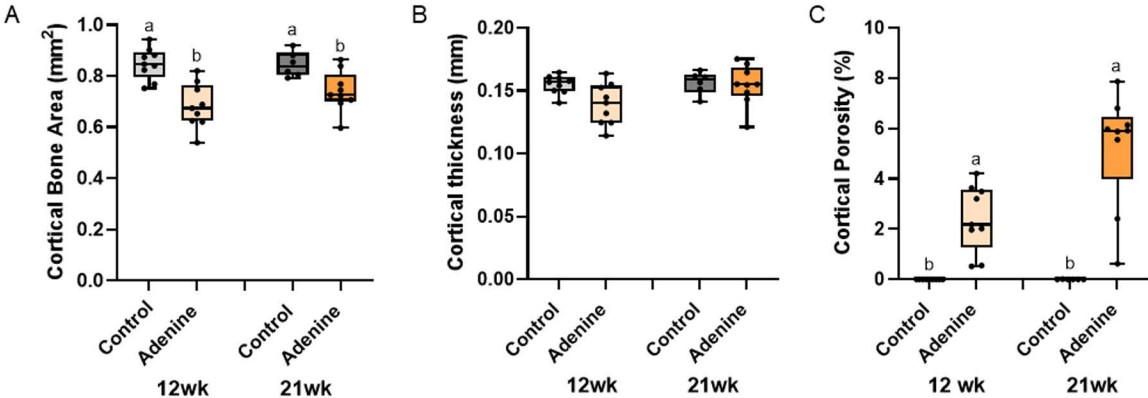

**Fig 4. Cortical microarchitecture of the femur.** A) Cortical bone area was lower in both adenine groups compared to both control groups with no impact of time. B) Cortical thickness was not different due to disease or time. C) Cortical porosity was higher in both adenine groups vs. both control groups. Bars not sharing the same letter are statistically different (p < 0.05).

but the relationship was inverse to 12-week with higher serum PTH associated with a lower %RANKL+ staining in cortical osteocytes.

## Discussion

This study demonstrated that 12-weeks of adenine-induced CKD with high circulating PTH resulted in high bone formation rate and elevated PTHR1 and RANKL in osteocytes. Most notably we show that despite 9 more weeks of compromised kidney function and continued elevated PTH, there was a blunted skeletal response in bone formation rate and osteocytes positive for PTHR1 and RANKL. Regression analyses highlight positive relationships between serum PTH and cortical osteocyte PTHR1 and RANKL at the 12-week time point which is lost by the 21-week time point. These results show the time-dependent PTH dynamics in CKD and suggest that prolonged exposure to high PTH alters the osteocyte response to PTH. This has implications both for using serum PTH as a sole surrogate for bone status in CKD as well as the timing and likely efficacy of therapeutics aimed to reduce high circulating PTH to protect bone.

The clinical progression of CKD is typically slow and heterogeneous over months or years. In contrast, animal models of CKD often progress quite rapidly and homogenously. Our previous work with the Cy/+ genetic model of CKD demonstrates a phenotype that relates to clinical CKD-MBD, but the disease progresses rapidly at later stages [16,26] not allowing for exploration of a more steady-state disease phenotype. In our previous experience with the adenine-induced CKD model in C57Bl/6J mice, we have noted what appears to be a more stable disease condition between 10–14 weeks post-induction of adenine [27,28]. Therefore, in this study we aimed to explore disease stability and skeletal properties over a more prolonged period. We found the overall well-being of the animals to be high, as evidenced by the fact that adenine-induced CKD mice were active, well-groomed, displaying natural nest building behaviors, and had no additional weight loss between 12- and 21-weeks. BUN was high at both timepoints indicating the presence of kidney disease, while PTH was statistically higher at 21-weeks vs. 12-weeks. Overall, the mice exhibited a maintained CKD state with continued elevations in PTH.

Cortical porosity is a common skeletal feature of CKD with high PTH. Previously, we have demonstrated the progression of porosity development in different models. In the naturally progressing Cy/+ rat model, cortical porosity develops rapidly within ~5 weeks in the tibia [16] and appears throughout the skeleton by ~35 weeks of age [29]. In the development of adenine-induced CKD, C57Bl/6J mice had approximately double the cortical porosity in the femur between 6- and 10-weeks post adenine induction [17]. In the current study, we found that cortical porosity was present in both adenine

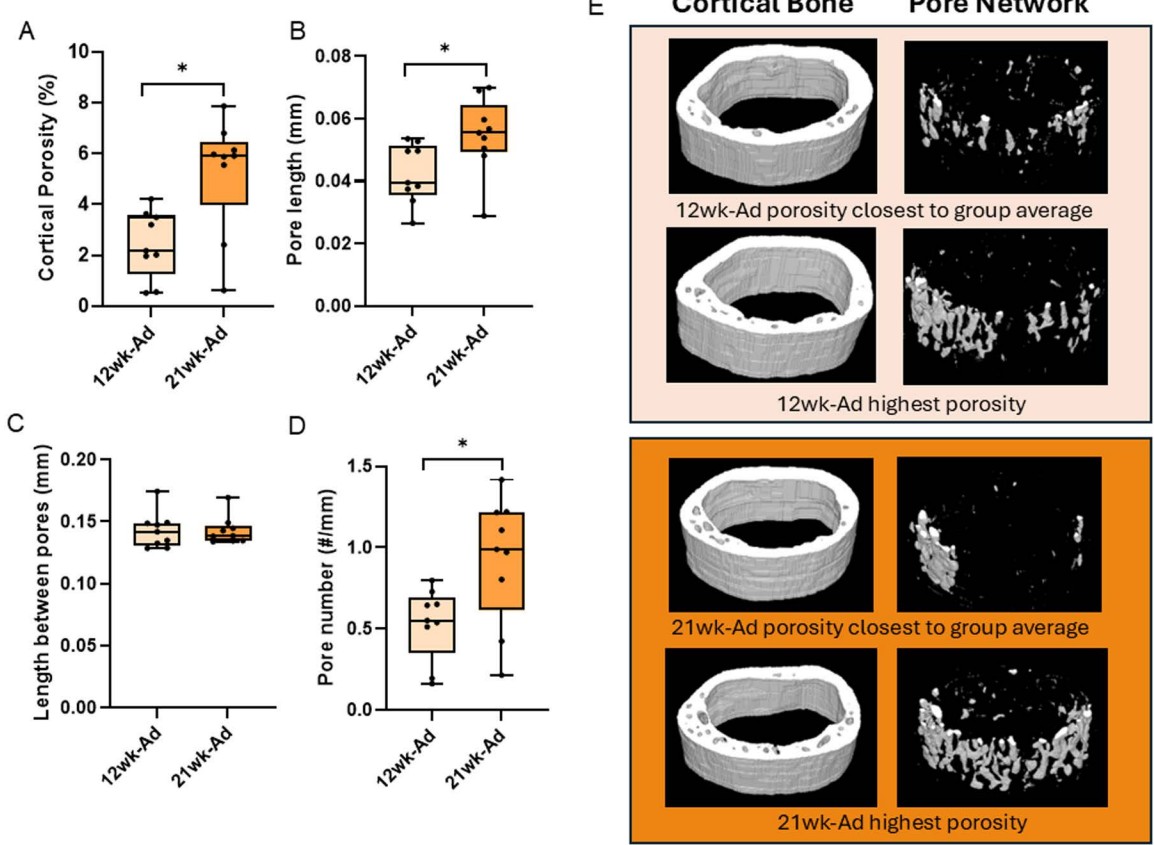

**Fig 5. Volumetric analysis of pore network from the midshaft femur in 12-week and 21-week adenine mice.** A) Volumetric cortical porosity was higher in 21-week adenine vs. 12-week adenine. B) Pore length was greater in 21-week adenine vs. 12-week adenine. C) There were no differences between groups in the average length between pores. D) Pore number was higher in 21-week adenine vs. 12-week adenine. E) Representative images of volumetric cortical bone area (left) and pore network (right) in 12- and 21-week adenine mice. *Indicates statistical difference between groups from t-test (p < 0.05).

groups regardless of time. Analysis of the volumetric pore network showed that 21-week adenine mice had greater pore number than did the 12-week adenine mice. We do not know at what point in the 9-week period between the groups the development of porosity was most rapid, if it continued to develop at a steady state, or if porosity development slowed at some point. Intracortical bone formation rate analyses and osteoclast assessments within the pores in adenine-CKD groups showed no difference between the two timepoints indicating the formation/resorption balance was not different at the two study endpoints.

An intriguing finding of this study was lower bone formation rate in the 21-week adenine-CKD mice vs. the 12-week adenine-CKD mice which occurred despite the continued high PTH. This was the result of both lower mineralized surface and lower mineral apposition rate at the later timepoint indicating both a reduction in the number and activity of osteoblast teams on trabecular surfaces. This counter-intuitive finding led us to investigate whether a hyporesponsiveness to PTH occurred due to the prolonged exposure to high PTH. Immunohistochemistry for the PTH receptor, PTHR1, showed ~2-fold higher trabecular osteocytes positive for the receptor in 12-week adenine-induced CKD mice compared to controls; within the 21-week groups, adenine-CKD mice had higher (~50%) but more modest elevations compared to controls. Similar patterns were seen with RANKL, a key factor in bone upregulated by PTH. In cortical bone, there were no differences between 21-week control and 21-week adenine mice in either PTHR1- or RANKL-positive osteocytes. Regression

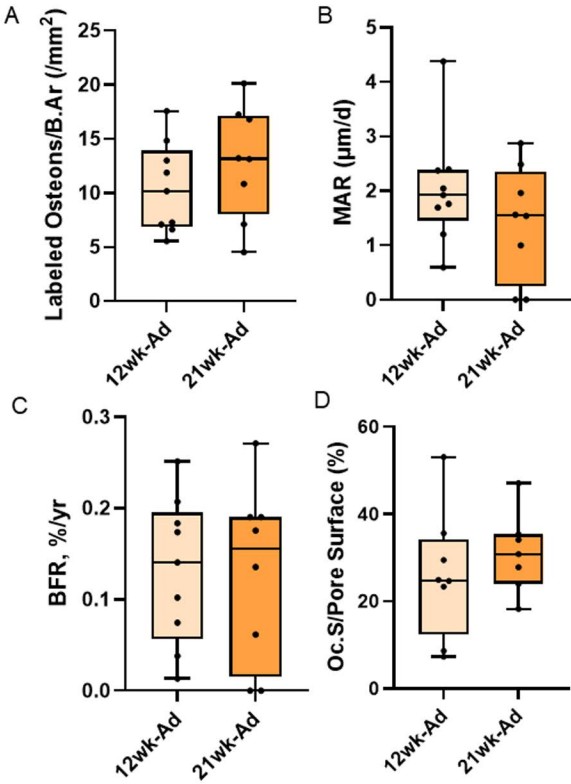

**Fig 6. Intracortical bone formation rate and osteoclast-covered pore surfaces in the femur of both adenine groups.** A) There was no difference in the number of labeled pores by bone area between adenine groups. B) Mineral apposition rate showed no statistical difference between groups. C) Intracortical bone formation rate was not different between 12-week and 21-week adenine mice. D) Osteoclast-covered pore surfaces were not different between adenine groups.

analyses demonstrated that circulating PTH statistically predicted ~60–70% of the variability in cortical osteocytes positive for PTHR1 and RANKL only at the 12-week timepoint. When regression analyses were performed with both timepoints together, there was no statistical relationship present. Furthermore, when only assessing the 21-week timepoint, there was no statistical relationship between PTH and PTHR1 and there was a mild, but statistically significant inverse relationship between PTH and RANKL. Together these data indicate that prolonged exposure to high PTH in the context of CKD eventually leads to altered osteocyte characteristics in response to elevated PTH.

We found that both osteoclast surfaces and osteoclast numbers in trabecular remained high in adenine-CKD mice regardless of the timepoint, despite lower bone formation rate at the later time point. PTH signaling increases bone turnover which typically results in high bone resorption and high bone formation. In this context, the high bone resorption remained at the later timepoint while bone formation rate was blunted. It is likely that various other factors contribute to high osteoclasts in CKD including circulating uremic toxins, oxidative stress, and a pro-inflammatory state induced by CKD. Therefore, in this model it appears that osteoclast drive remains high despite lower PTHR1 and reduced RANKL in osteocytes. One potential contributing factor to the high osteoclasts at both timepoints is increased osteocyte apoptosis as seen by elevated annexin V-positive osteocytes found in adenine-CKD mice at both timepoints, particularly in trabecular bone. Previously, we found lower annexin V-positive osteocytes in younger female C57BL/6J adenine-CKD mice at earlier time points [17] which we attributed to elevated PTH inhibiting osteocyte apoptosis. However, in this study at later timepoints in skeletally mature male mice, annexin V-positive osteocytes were higher in CKD. It is possible that a PTH-induced

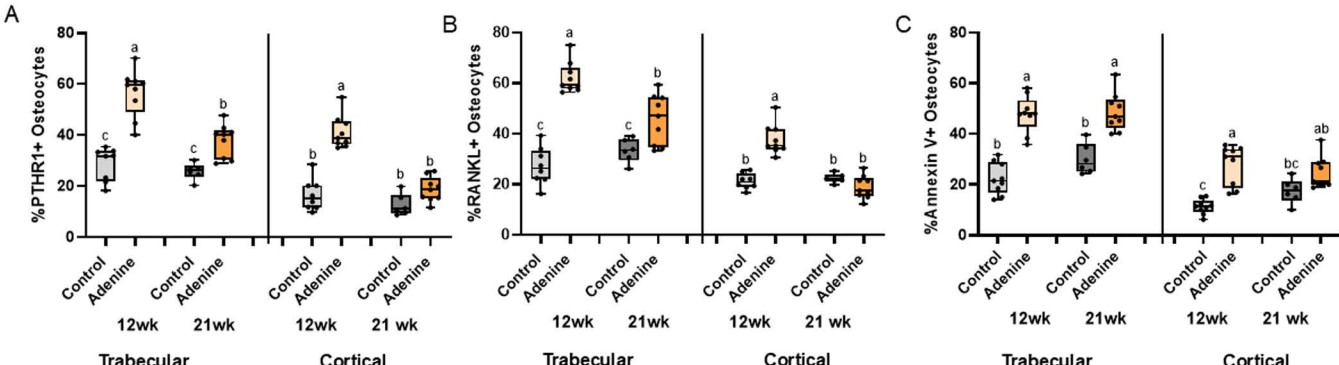

**Fig 7. Immunohistochemistry for PTHR1, RANKL, and annexin V in the trabecular bone of the proximal femur and midshaft femur cortical bone.** A) %PTHR1-positive osteocytes were highest in 12-week adenine in trabecular bone followed by 21-week adenine with both control groups having the lowest group average. In the cortical bone, 12-week adenine was higher than all other groups. B) %RANKL-positive osteocytes were highest in 12-week adenine with 21-week adenine lower and the two control groups having the lowest values in trabecular bone. In cortical bone, 12-week adenine was higher than all other groups. C) %Annexin V-positive osteocytes were higher in both adenine groups compared to both control groups in trabecular bone. In cortical bone, both adenine groups were higher than 12-week control. Bars not sharing the same letter are statistically different (p < 0.05).

decrease in apoptosis may be lost earlier in the disease course particularly as other CKD-induced factors increase. Since osteocyte apoptosis is a driving signal for targeted bone resorption [30], it is possible the high osteoclasts at the later time-point with blunted PTHR1 and RANKL is due to the continued higher osteocyte apoptosis. Additionally, this demonstrates that PTH signaling alone is not the only driver of skeletal changes due to CKD.

The concept of skeletal hyporesponsiveness to PTH in CKD, also called skeletal resistance to PTH in some literature, is poorly understood, but has been alluded to as a factor contributing to the highly variable skeletal response to CKD seen clinically. Recently, Evenepoel and Jørgensen highlighted that the variability of responsiveness to PTH clinically compromises its viability as a biomarker of bone turnover [15]. Some evidence in clinical literature points to continually high PTH in CKD resulting in lower responsiveness. For example, infusion of PTH showed a lower calcemic response in CKD patients [31]. Another study found that 46 hours of infusion with PTH(1–34) resulted in increased serum calcium in control and low bone turnover hemodialysis patients, but no change in high turnover hemodialysis patients [32]. There also appears to be population differences in skeletal responsiveness to PTH with differences in Japanese vs. Belgian hemodialysis patients [33]. Within one cohort of CKD patients, PTH levels above the upper limit of normal were associated with adynamic bone, a condition with low resorption and low formation [14]. As PTH stimulates increased bone turnover, this effect is hypothesized to be due to a lack of responsiveness to PTH in these patients [14]. Studies have also shown downregulation of PTHR1 mRNA in osteoblasts in ESRD patients [34] while another study found higher PTHR1 in bone compared to healthy controls in earlier CKD stages (stages 2–3) while stage 5 CKD, when circulating intact PTH was highest, was not different from control levels [35]. Overall, these studies demonstrate the complexity of the clinical skeletal response to CKD and allude to high variability the responsiveness to PTH.

One challenge hindering a greater understanding of skeletal hyporesponsiveness to PTH in CKD is a lack of animal models showing this effect. Many animal models move rapidly through advanced stages of CKD limiting the ability to study a prolonged state of high PTH. Without manipulation, most animal models show a clear relationship between high PTH and high bone turnover. A couple of studies have aimed to experimentally explore the impact of CKD without high PTH in animal models. One such study assessed the calcemic response of infusion of PTH in rats with parathyroidectomy with or without CKD induced via 5/6 nephrectomy (Nx) and found a decreased calcemic response in Nx rats [36]. Another study assessed thyroparathyroidectomy (TPTx) and 5/6 Nx in rats with a constant infusion of a physiological dose of PTH and showed low bone formation rate and reduced expression of PTHR1 in rats with both TPTx and Nx compared to those with only

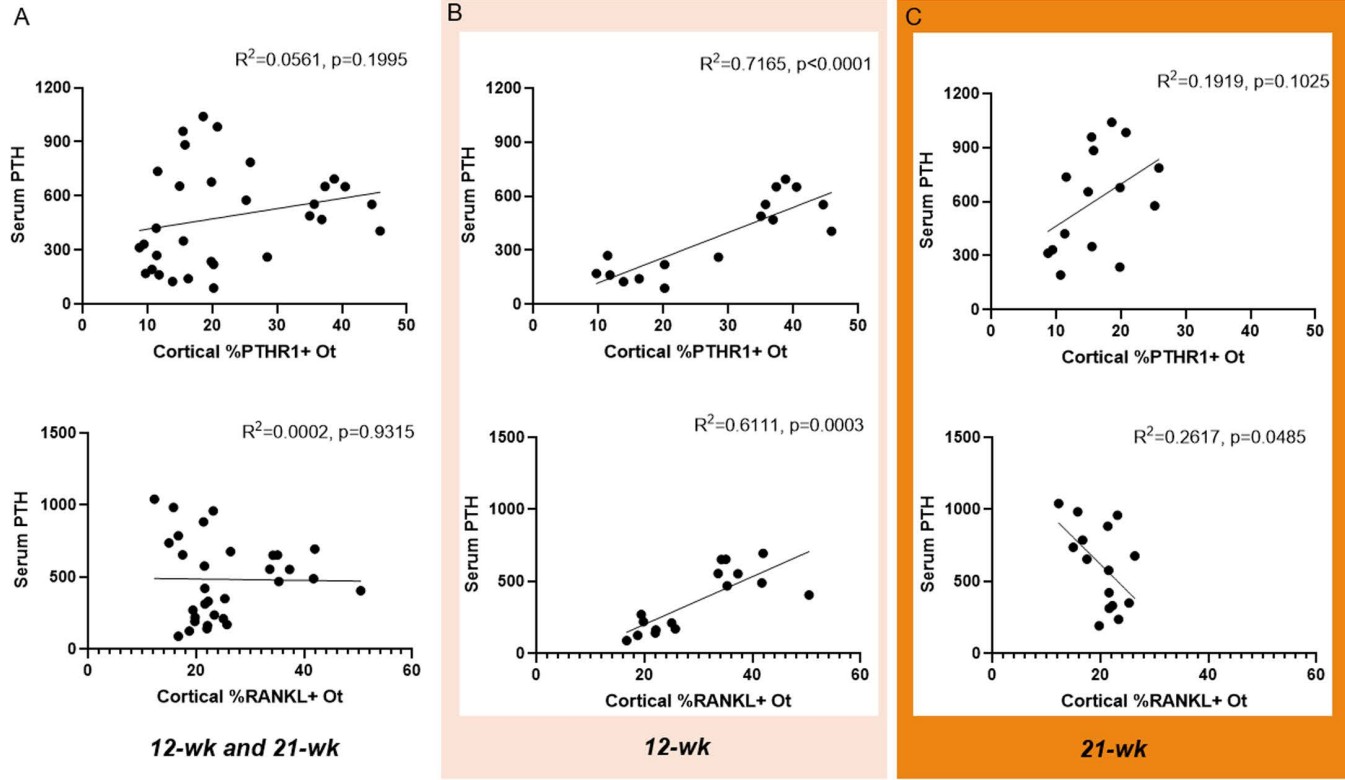

**Fig 8. Regression plots of serum PTH to cortical %positive osteocytes for PTHR1 and RANKL.** A) Top: full regression with both timepoints showed no relationship between serum PTH and cortical PTHR1. Bottom: full regression with both 12- and 21-week groups showed no relationship between serum PTH and cortical RANKL. B) Top: 12-week only regression showed statistical relationship between serum PTH and cortical PTHR1 ($R^2 = 0.7165$, p < 0.0001). Bottom: There was also a positive relationship ($R^2 = 0.6111$, p = 0.0003) between serum PTH and cortical RANKL at 12-weeks. C) Top: Within the 21-week groups, there was no relationship between serum PTH and cortical PTHR1. Bottom: At the 21-week timepoint, serum PTH predicted ~26% of variability in cortical RANKL ($R^2 = 0.2617$, p = 0.0485).

TPTx [37]. Both studies indicate that CKD itself may alter bone's responsiveness to PTH, but this effect may be masked by naturally progressing secondary hyperparathyroidism. In our current study, we found blunted bone formation rate and lower PTHR1 and RANKL in adenine-CKD mice with no manipulation other than length of time with adenine-induced CKD. This study indicates that the length of time exposed to high PTH likely plays a role in the development of skeletal hyporesponsiveness to CKD. As the adenine model requires no further experimental manipulation, it potentially could lead to further insights into the relationship between time, secondary hyperparathyroidism, and the skeletal phenotype in CKD.

Limitations of this study include using only male mice. We previously have reported similar phenotypes in adenine-induced CKD in both male and female C57Bl/6J mice [19,38], however, we have not assessed female mice to a timepoint later than 14-weeks post-induction with CKD [28]. Furthermore, we only assessed one age and are unable to conclude how this response could be different at different ages. Other factors associated with CKD including systemic inflammation and oxidative stress may also contribute to the skeletal response to CKD, but the causal relationships have not yet been defined. Finally, some *in vitro* studies have alluded to how uremic toxins may contribute to inducing skeletal hyporesponsiveness to CKD. For example, uremic ultrafiltrate from hemodialysis patients reduced PTHR1 mRNA when cultured with osteoblast-like cells [39] while another study found the uremic toxin, indoxyl sulfate, reduced PTHR1 expression in cultured osteoblasts [40]. Future work should assess uremic toxins over time and compare with circulating PTH and the skeletal phenotype.

In conclusion, this study shows that 12-weeks of adenine-induced CKD resulted in high bone formation rate and a high percentage of osteocytes positive for PTHR1 and RANKL; however, nine weeks more of exposure to high PTH led to blunted responses in all these factors. These data allude to a time-based development of hyporesponsiveness to chronically elevated PTH in CKD. Whether this is due solely to the prolonged exposure to PTH or other CKD-related factors or a combination of both is unclear; however, this demonstrates that measurements of serum PTH in CKD may not correlate with skeletal changes. Furthermore, these data point to the need to greater understand the timing and how to assess the efficacy of therapeutics that lower PTH in the time course of CKD development to maximize the effectiveness of these treatments on bone.

## Supporting information

**S1 Fig. Nesting scores and body weights during adenine induction and immediate recovery.** A) Cage nesting scores recorded during weeks 3–12 of the study. B) Body weights from baseline through week 12 of the study. *Indicates difference in body weight between control and adenine beginning after one week on the adenine diet.
(TIF)

**S2 Appendix. Raw data points for datasets.**
(XLSX)

## Author contributions

**Conceptualization:** Corinne E. Metzger, Matthew R. Allen.

**Formal analysis:** Corinne E. Metzger.

**Funding acquisition:** Matthew R. Allen.

**Investigation:** Corinne E. Metzger, Landon Y. Tak, Samantha Scholz.

**Project administration:** Corinne E. Metzger.

**Resources:** Matthew R. Allen.

**Supervision:** Matthew R. Allen.

**Visualization:** Corinne E. Metzger.

**Writing – original draft:** Corinne E. Metzger.

**Writing – review & editing:** Landon Y. Tak, Samantha Scholz, Matthew R. Allen.

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
