## [Decision Letter · Decision Letter 0]

27 Feb 2025

PONE-D-25-04756Prolonged secondary hyperparathyroidism in adenine-induced CKD leads to skeletal changes consistent with skeletal hyporesponsiveness to PTHPLOS ONE

Dear Dr. Metzger,

Thank you for submitting your manuscript to PLOS ONE. After careful consideration, we feel that it has merit but does not fully meet PLOS ONE’s publication criteria as it currently stands. Therefore, we invite you to submit a revised version of the manuscript that addresses the points raised during the review process.

We look forward to receiving your revised manuscript.

Kind regards,

Ewa Tomaszewska, DVM Ph.D

Academic Editor

PLOS ONE

Journal Requirements:

This study was funded by the Department of Veterans Affairs via grants I01BX003025 and IK6-BX006479 awarded to MRA. Histological processing was completed through the Histology Lab Core Service in the Indiana Center for Musculoskeletal Health (P30-AR083854 to AG Robling).

This study was funded by the Department of Veterans Affairs via grants I01BX003025 and IK6-BX006479 awarded to MRA.

Reviewers' comments:

Reviewer's Responses to Questions

**Comments to the Author**

1. Is the manuscript technically sound, and do the data support the conclusions?

Reviewer #1: Partly

Reviewer #2: Yes

2. Has the statistical analysis been performed appropriately and rigorously? 

Reviewer #1: Yes

Reviewer #2: Yes

3. Have the authors made all data underlying the findings in their manuscript fully available?

Reviewer #1: Yes

Reviewer #2: Yes

4. Is the manuscript presented in an intelligible fashion and written in standard English?

Reviewer #1: Yes

Reviewer #2: Yes

5. Review Comments to the Author

Reviewer #1: The longitudinal design of the study is a notable strength, as it captures the progression of skeletal changes over time—a feature often lacking in animal models of CKD.

However, there are several areas where the study could be improved. First, the exclusive use of male mice limits the generalizability of the findings, given the well-documented sex-specific differences in bone metabolism and CKD progression. Including female mice in future studies would provide a more comprehensive understanding of these dynamics. Second, while the study identifies altered osteocyte characteristics and reduced expression of PTH receptor 1 (PTHR1) and receptor activator of nuclear factor κB ligand (RANKL) as potential mechanisms, it does not explore the role of other CKD-related factors, such as uremic toxins or systemic inflammation, which may also contribute to skeletal hypo responsiveness.

Reviewer #2: The manuscript entitled "Prolonged secondary hyperparathyroidism in adenine-induced CKD leads to skeletal

changes consistent with skeletal hyporesponsiveness to PTH" have designed and written well. No any major correction and grammatical mistake reported at present as per my view. The present manuscript can be accepted for publication.

6. PLOS authors have the option to publish the peer review history of their article (what does this mean?). If published, this will include your full peer review and any attached files.

Reviewer #1: **Yes: **Dr.Zena A.A.Hadedy

Reviewer #2: No

---

## [Author Response · Author response to Decision Letter 1]

6 Mar 2025

Reviewer #1: The longitudinal design of the study is a notable strength, as it captures the progression of skeletal changes over time—a feature often lacking in animal models of CKD.

However, there are several areas where the study could be improved. First, the exclusive use of male mice limits the generalizability of the findings, given the well-documented sex-specific differences in bone metabolism and CKD progression. Including female mice in future studies would provide a more comprehensive understanding of these dynamics. Second, while the study identifies altered osteocyte characteristics and reduced expression of PTH receptor 1 (PTHR1) and receptor activator of nuclear factor κB ligand (RANKL) as potential mechanisms, it does not explore the role of other CKD-related factors, such as uremic toxins or systemic inflammation, which may also contribute to skeletal hypo responsiveness.

Response: Thank you for your comments. In the limitations paragraph we addressed the limitation of only assessing male mice as well as the need to address uremic toxins and their role in future studies with citations of in vitro work on that front. We have now also included a sentence addressing the unexplored role of other factors like systemic inflammation and oxidative stress.

Reviewer #2: The manuscript entitled "Prolonged secondary hyperparathyroidism in adenine-induced CKD leads to skeletal

changes consistent with skeletal hyporesponsiveness to PTH" have designed and written well. No any major correction and grammatical mistake reported at present as per my view. The present manuscript can be accepted for publication.

Response: Thank you for your review.

---

## [Editor Report · Decision Letter 1]

29 Apr 2025

Prolonged secondary hyperparathyroidism in adenine-induced CKD leads to skeletal changes consistent with skeletal hyporesponsiveness to PTH

PONE-D-25-04756R1

Dear Dr. Corinne Metzger,

We’re pleased to inform you that your manuscript has been judged scientifically suitable for publication and will be formally accepted for publication once it meets all outstanding technical requirements.

Kind regards,

Ewa Tomaszewska, DVM Ph.D

Academic Editor

PLOS ONE

Additional Editor Comments (optional):

Dear Authors,

Thank you for the submitted revisions. I confirm that you have appropriately addressed the reviewer’s comments, including the limitation of using only male mice and the potential roles of systemic inflammation, oxidative stress, and uremic toxins. These points have been properly incorporated into the revised manuscript.

with best regards

Ewa Tomaszewska
---

## [Editor Report · Acceptance letter]

PONE-D-25-04756R1

PLOS ONE

Dear Dr. Metzger,

I'm pleased to inform you that your manuscript has been deemed suitable for publication in PLOS ONE. Congratulations! Your manuscript is now being handed over to our production team.

Kind regards,

on behalf of

Professor Ewa Tomaszewska

Academic Editor

PLOS ONE